# Oxidative Stress and Cocaine Intoxication as Start Points in the Pathology of Cocaine-Induced Cardiotoxicity

**DOI:** 10.3390/toxics9120317

**Published:** 2021-11-24

**Authors:** Ekaterina Georgieva, Yanka Karamalakova, Radostina Miteva, Hristo Abrashev, Galina Nikolova

**Affiliations:** 1Department of General and Clinical Pathology, Forensic Medicine, Deontology and Dermatovenerology, Medical Faculty, Trakia University, 11 Armeiska Str., 6000 Stara Zagora, Bulgaria; ekaterina.georgieva@trakia-uni.bg (E.G.); radostina.dimitrova@trakia-uni.bg (R.M.); 2Department of Medical Chemistry and Biochemistry, Medical Faculty, Trakia University, 11 Armeiska Str., 6000 Stara Zagora, Bulgaria; yanka.karamalakova@trakia-uni.bg; 3Department of Vascular Surgery, Medical Faculty, Trakia University, 11 Armeiska Str., 6000 Stara Zagora, Bulgaria; hristo.abrashev@trakia-uni.bg

**Keywords:** cocaine, cardiotoxicity, nitroxide radicals, oxidative stress, sudden cardiac death

## Abstract

Psychomotor stimulants are the most commonly used prohibited substances after cannabis. Globally, their use reaches epidemiological proportions and is one of the most common causes of death in many countries. The use of illicit drugs has negative effects on the cardiovascular system and is one of the causes of serious cardiovascular pathologies, ranging from abnormal heart rhythms to heart attacks and sudden cardiac death. The reactive oxygen species generation, toxic metabolites formation, and oxidative stress play a significant role in cocaine-induced cardiotoxicity. The aim of the present review is to assess acute and chronic cocaine toxicity by focusing on the published literature regarding oxidative stress levels. Hypothetically, this study can serve as a basis for developing a rapid and effective method for determining oxidative stress levels by monitoring changes in the redox status of patients with cocaine intoxication.

## 1. Introduction

Cocaine, together with amphetamine and its analogs, belongs to the group of psychomotor stimulants with different pharmacological and pathophysiological properties and with a well-described synergy between them [1,2]. The effect of various drugs on dopamine regulation and the role of the dopamine molecule in the development of drug dependence and addiction have been well studied [3]. As a central nervous system (CNS) stimulant, cocaine causes a feeling of euphoria, increased motor activity, and a reduced feeling of fatigue and hunger [4,5]. In response, it leads to addiction and dependence, leading to chronic use, and increases the risk of damage to the whole organism [6]. In addition to its effects on the CNS, its use can lead to significant cardiovascular complications and sudden cardiac death (SCD). SCD is characterized as a dynamic event and represents the sudden and unexpected death of cardiac etiology. It is defined as a sudden, unexpected pulseless condition, which usually occurs within ≤1 h of the onset of symptoms and is often due to cardiac arrhythmia [7,8,9,10].

Depending on the dose and the presence of concomitant cardiovascular disease, the clinical effects vary widely and include vasoconstriction, arrhythmia, tachycardia, aortic dissection, heart attack, myocardial ischemia, etc. [11,12]. Not only are the harmful effects of cocaine well-known, but also those of its metabolites [13], which can cause serious and irreversible damage to the entire cardiovascular system [14,15]. Cocaine exerts its toxicity on the human body by generating reactive oxygen species (ROS) such as hydrogen peroxide, hydroperoxides, alkyl peroxides, superoxide, hydroxyl, and others [16,17]. According to the data, ROS production and oxidative damage are considered significant factors in cocaine-induced cardiotoxicity [18,19] and the pathogenesis of cardiovascular disease [20].

In this review, we examined the detrimental effects of cocaine and its metabolites on the cardiovascular system. Moreover, we accented to the role of oxidative stress (OS) in the pathology of cocaine-induced cardiotoxicity. The review aims to develop methods for diagnosing acute and chronic cocaine toxicity by determining OS levels in biological samples used mainly in emergency medicine and toxicology clinics.

## 2. Materials and Methods

### 2.1. Search of the Literature, Materials, and Methods

The aim of our study was to examine the role of oxidative stress in cocaine-induced cardiomyopathy. A preliminary review of the literature was conducted, with a focus on the involvement of free radicals in the mechanisms of induction of pathological changes in the cardiovascular system in cocaine users. In accordance with the requirements for systematic reviews and meta-analysis protocols (PRISMA-P), a topic and a research question were identified. Detailed studies were performed in the field of cocaine intoxication and cocaine metabolites directly related to the cardiovascular system and cocaine-induced cardiac pathologies. A comprehensive literature search strategy was developed according to the chosen topic. To ensure the accuracy and completeness of this systematic review, we searched for relevant scientific articles in Scopus, Web of Science, Medline, HealthCare, PubMed, PMC Europe, and Research Gate. An in-depth analysis of the English scientific literature in the period from 1976 to 2021 was conducted, which contains in its titles the keywords “drug abuse”, “intoxication”, “cocaine”, “metabolites of cocaine”, “neurotransmitters”, “dopamine receptors”, “cardiotoxicity of cocaine”, “sudden cardiac death”, “oxidative stress”, “free radicals”, “oxidative damages”, and managed to include studies according to PRISMA guidelines [21]. Over 550 clinical, laboratory, epidemiological studies and literature reviews related to stimulant use and cocaine-induced cardiotoxicity were studied, from which we selected 137 articles directly related to our topic (Figure 1).

### 2.2. Inclusion Criteria

To fulfill the aim of the review, we included articles from the scientific literature and reviews in English that focused on: (i) effect of cocaine on the central nervous system; (ii) effect of cocaine and metabolites on the cardiovascular system; (iii) cocaine-related sudden cardiac death; (iv) free radicals, oxidative stress and toxicity of cocaine.

### 2.3. Exclusion Criteria

We did not include historical reviews and articles that are not in English and are not directly related to cocaine-induced pathological changes in the cardiovascular system. Scientific publications describing the effects of various narcotic substances (amphetamine, methamphetamine, heroin, methadone, marijuana, etc.) were not considered. The toxic effects of cocaine and its metabolites on other organs and systems, such as respiratory, immune, and digestive, were not considered as they are not the subject of this topic.

## 3. Effect of Cocaine on Central Neural System and Cardiovascular System

### 3.1. Cocaine Action on the Brain System

Cocaine belongs to the group of psychomotor stimulants and is considered one of the most addictive drugs. Its narcotic effect consists of increasing extracellular dopamine (DA), and dopamine dysregulation underlies the addictive behavior [22,23]. The mechanism involves inhibition of dopamine reuptake in the central neural system (CNS) [11,24]. The cocaine molecule acts on the presynaptic transporters of monoamines and facilitates the activity of the monoamine neurotransmitters dopamine, norepinephrine (NE), and serotonin (SE) in the CNS and peripheral nervous system [1,25,26]. Depending on the stimulant, the pathophysiological mechanisms of drug action include direct toxicity, neurohormonal activation, altered calcium homeostasis, and oxidative stress. There are three main brain systems involved in drug reward—dopamine, opioid, and γ-aminobutyric acid/aminobutyric acid receptor (γ-GABA/GABA). Therefore, reward systems for the brain have a multidetermined neuropharmacological basis with separate neurochemical processes involving different, albeit overlapping, neuroanatomical patterns [27,28]. Experimental data show that the Nucleus accumbens septi (limbic region) and the dorsal caudal nucleus are the two main target areas in CSN in which increases in extracellular concentrations of DA are observed [29]. In addition to the limbic region, stimulants exert their effects on the brain by increasing the release of norepinephrine in the prefrontal cortex [30].

### 3.2. Dopamine Transporter—The Main Target of Stimulants

The main target of cocaine in the human body is the dopamine transporter (DAT). Cocaine blocks the action of the transporter, preventing the DA absorption in the CNS and thus allowing its accumulation at the synapse. Initially, the pharmacological effect it exerted was DAT blockade, followed by an increase in DA concentration and dopamine receptors activation [31,32]. Through a synaptic mechanism, the drug directly amplifies the mesolimbic dopaminergic signal at the DA receptor, increasing synaptic dopaminergic concentrations and thus mediating certain behavioral effects [33]. Dopamine accumulates in the synaptic cleft and causes increased activity of postsynaptic receptors, eliciting an enhanced response in the host cell (Figure 1) [34]. In cocaine users, catecholamines can be increased up to five times [11]. Retention of the cocaine molecule by DAT leads to overstimulation of dopaminergic neurons and excessive synaptic metabolism of the neurotransmitter [35]. Increasing the DA absolute concentration at the synapse and the time interval in which the neurotransmitter remains at the site of the postsynaptic receptor disturbs the balance between the release of DA and the reuptake of the dopamine molecule [33]. The sympathomimetic effects of cocaine are manifested not only by binding the cocaine molecule to the dopamine transporter but also to the serotonin and norepinephrine transporter (NET), which increases sympathetic stimulation and causes vasoconstriction, hypertension, and increased myocardial oxygen demand [36]. Cocaine inhibits the reuptake of the monoamine neurotransmitter 5-hydroxytryptamine (5-HT) by blocking the action of SERTPR, enhancing the secretion of adrenaline and noradrenaline from the adrenal cortex, which enhances the effect of norepinephrine. As a result, sustained supraphysiological extracellular levels of various catecholamines were observed [37], which defines the drug as a potent sympathomimetic agent with a direct cardiotoxic effect [38].

Especially important for the drug’s amplified effects is the way they are administered (intravenous, intranasal, and smoking). Volkow et al. determine the rate at which cocaine enters the brain as a key parameter in its effectiveness in blocking DAT. Using 11C-labeled cocaine and positron emission tomography (PET), they found a significant blockade of the dopamine transporter in all modes of cocaine administration. A dose-dependent effect was observed with intravenous and intranasal administration but not with cocaine smoking [39]. Its rapid absorption by the brain leads to rapid changes in DA levels. Moreover, for the enhancing properties of the drug, the involvement of a “phasic” dopamine firing is particularly important, which is characterized by sharp fluctuations in neurotransmitter levels [40]. Similar to cocaine, amphetamine (AMPH) acts on DAT by activating the mesolimbic dopaminergic pathway. The mechanism involves an increase in extracellular dopamine and prolongation of DA-receptor signaling in the striatum [41,42]. Both cocaine and amphetamines increase the time interval in which DA remains at the postsynaptic receptor [33].

Cocaine works mainly by blocking the dopamine transporter, while amphetamine competitively prevents the reuptake of dopamine by the DAT. It is assumed that the action of methylphenylamine also depends on its concentration; furthermore, the dose is crucial for determining the effects of abuse of amphetamine and/or its analogs. Thus, at low concentrations, AMPH acts primarily as a DAT blocker, while high concentrations increase and promote DAT-mediated back transport of dopamine from the cytoplasm to the synaptic cleft [22,43,44].

## 4. Cocaine-Related Sudden Cardiac Death

The heart muscle is one of the main targets of many drugs and various chemicals, and cardiovascular diseases (CVDs) are the leading cause of death worldwide. Various cardiovascular pathologies are directly dependent on constant drug intake [45] and characterized by complex heterogeneous pathophysiological mechanisms [46]. Stimulant abuse is a major cause of new or exacerbation of pre-existing cardiac pathology, which increases morbidity and mortality in general [47]. Cocaine has very powerful direct effects on cell membranes by blocking sodium channel activity, so cocaine intoxication is a common cause of various cardiovascular events. Cardiotoxic effects are associated with the development of cardiac contractile dysfunction, high blood pressure [11], tachycardia, arrhythmia [12,28] and sudden cardiac death (SCD) [48,49]. Its use can lead to myocardial infarction, causing vasoconstriction of the coronary artery [50,51]. Hemorrhagic stroke, infective endocarditis, myocarditis, acute pulmonary edema, and aortic dissection are common side effects associated with hypertension in cocaine addicts [52]. The sudden rise in blood pressure and weakening of the vessel wall leads to local dilation of blood vessels (arteries or veins), which can cause aneurysms [53,54]. Smoking crack or cocaine causes drug accumulation in the heart tissue and causes acute cardiotoxicity, directly affecting the myocardium [55,56]. The pathophysiology of cocaine-related myocardial ischemia and infarction involves one or a combination of several factors. On the one hand, it increases heart rate, blood pressure, and contractility increases the need for oxygen in the myocardium. On the other hand, the supply of oxygen to the myocardium is insufficient, resulting in its balance in the myocardial tissue being disturbed [52]. Another mechanism characteristic of cocaine cardiotoxicity involves conduction disturbances [57]. Blocking sodium channels and increasing calcium flow with a subsequent vasoconstrictor reaction is thought to be one of the leading mechanisms of cocaine cardiotoxicity [58]. By affecting myocardial electrical impulses, there is an increase in the contractility and electrical conductivity of cardiomyocytes [11], as well as a change in cardiac conduction [52]. At low doses, cocaine blocks sodium channels and produces enhanced sympathomimetic activity. Due to its direct effects on cardiac ion channels, inhibition of L-type calcium current Ca^2+^ delayed rectifier potassium (K) currents, and a sodium (Na) current in cardiomyocytes was observed. This leads to increased PR, QRS, and QT intervals and disruption of the coordinated electrical activity of the heart [59]. Inactivation of the sodium and potassium channels by cocaine results in decreased myocardial contractility, intracardiac conduction delay, and myocardial suppression. Prolongation of the cardiac ventricular depolarization period led to reentrant arrhythmia and reduced left ventricular ejection fraction [38]. Thus, the drug exerts direct toxicity on the heart muscle, which is associated with arrhythmias, due to secondary blockade of sodium channels and elevated levels of norepinephrine [60]. Besides, circulating catecholamines affect the coronary vasculature as vasoconstrictors and can cause ischemia and myocardial infarction (MI) [61]. Cocaine use has also been associated with a large rapid and transient increase in the risk of acute myocardial infarction in patients who are otherwise at relatively low risk [62]. Chronic cocaine use causes irreversible structural damage to the heart, including arterial endothelial damage, vascular damage, and progression of atherosclerosis [63]. For individuals with an initially low risk of atherosclerosis, cocaine use led to sudden death in 76% of cases. The mechanism involves thrombus formation due to increased platelet aggregation at sites of atherosclerotic narrowing and enhanced coronary arterial vasoconstriction [64]. Therefore, chronic drug use and subsequent left ventricular hypertrophy are considered to be the leading cause of myocarditis, dilated cardiomyopathy, and heart failure [6]. About 43% of emergency department visits after drug intoxication are due to cocaine use, with the highest percentage of men between 35 and 44 years of age [65]. Mortality associated with cocaine use is also common in 30-year-old men and is most common at home and on the weekends [66].

## 5. Metabolites of Cocaine with Expressed Cardiotoxicity

Cocaine is metabolized mainly in two different ways. The main chemical reactions to cocaine that produce toxic metabolites include hydrolysis and decarboxylation. Although there is individual variability, cocaine can be detected in oral fluids for up to 24 h in urine for approximately four days, and in hair, especially in chronic use, for up to 90 days [12].

Cocaine undergoes a rapid metabolism, producing two main hydrophilic metabolites—benzoylegonine and ecgonine methyl ester. The metabolites also involved norcocaine, norcocaine nitroxide, N-hydroxynorcocaine, norcocaine nitrosonium, and cocaine iminium [17]. Besides, highly toxic formaldehyde is formed at a significant rate [67]. Benzoylecgonine and ecgonine methyl ester are the two major metabolites known to cause hypertension [38]. It can be detected in the urine for 1 to 2 days after intravenous administration of 20 mg of cocaine and 2 to 3 days at a higher dose administered intranasally. In chronic users, the maximum time for detection of benzoylecgonine in urine is 22 days after the last use [68]. Norcocaine, a metabolite known for its vasoconstrictive effects, was also shown to be highly cardiotoxic [69,70]. Intravenous administration of norcocaine to rats at a dose of 1 mg/kg caused a decrease in heart rate and an increase in plasma adrenaline levels. This suggests that cocaine, similar to norcocaine, may be implicated in cocaine cardiovascular toxicity [20,71]. Ecgonine can be detected 98 h after administration of cocaine in the urine [72] and up to 8 days after a single dose [73].

Countless interactions were reported between cocaine, alcohol, and other substances. The drug transesterification in the presence of ethanol (EtOH) results in the formation of a potent pharmacologically active metabolite cocaethylene (ethylbenzoylecgonine) (Figure 2) [74,75], which shows direct cardiotoxicity, with a proven high risk of heart attack [76]. Ethanol enhances and prolongs the effects of cocaine on the cardiovascular system, and the combination of cocaine and ethanol is more cardiotoxic than any other. Their combination significantly increases heart rate and can lead to myocardial depression, decreased coronary arterial blood flow, dysrhythmias, and sudden cardiac death, probably due to cocaethylene toxicity [77]. Concomitant use of cocaine and nicotine significantly exacerbates the harmful effects of the drug on the heart by disrupting the supply of oxygen to the myocardium by potentiating coronary arterial vasoconstriction [78]. 

Cocaine was reported as a strong vasoconstrictor and can be detected in the blood and urine even 4 days after use. The time to peak blood concentration varies and depends on the route of administration (smoking, oral administration, nasal inhalation, and intravenous injection). When smoking or intravenously injected, the maximum concentration of the drug is reached between 1 and 5 min, and orally between 60 and 90 min [79]. The time to peak subjective effect of cocaine averaged 14.6 min after insufflation compared with 3.1 min after injection [2]. he duration of pharmacological action also varies depending on the method of administration, lasting from 5 to 60 min when administered intravenously [60]. The maximum concentration of cocaine in the blood causes temporary vasoconstriction, and after 60 min, the diameter of the coronary artery returns to its original state. In cases of intranasal cocaine administration, recurrent vasoconstriction lasting up to 90 min was observed due to the metabolites benzoylecgonine and ethylmethyl ecgonine [80].

## 6. Acute and Chronic Effects of Cocaine Cardiovascular Toxicity and Pathological Changes in the Cardiovascular System

In general, the number of cardiovascular events following cocaine abuse is increasing dramatically, but distinguishing the effects and determining the consequences in humans is troublesome, as there are a number of factors such as differences in administration, doses, mixing cocaine with other drugs (alcohol, caffeine, and amphetamines), the presence of contaminants of the probes (procainamide, quinidine, and antihistamines), etc. [81]. Numerous and varied pathological changes are known after acute and chronic cocaine abuse, such as myocardial infarction, myocarditis, catecholamine-induced cardiomyopathy, chronic cardiomyopathy, etc., but their clinical manifestations overlap to a large extent [82]. In order to determine the cause and mechanism of sudden cardiac death (SCD), it is necessary for a detailed forensic investigation to be conducted, which allows for the understanding of the intimate mechanism and whether the disease is due to acute or chronic cocaine use. Toxicological analyses would provide clarity regarding the toxic agent, while macroscopic and histopathological examination would make it possible to establish the mechanism and role of the intoxicant that led to the lethal outcome. Mechanisms associated with an increased risk of SCD after acute or chronic cocaine use include coronary vasoconstriction, accelerated atherosclerosis, increased thrombogenesis, left ventricular hypertrophy, increased myocardium oxygen needs, acute myocardial infarction, etc. [83].

### 6.1. Acute Toxicity of Cocaine. Myocardial Infarction, Arrhythmias, Sudden Cardiac Death

Cocaine-induced cardiotoxicity can cause a variety of structural and functional damage to cardiac tissue [48]. Acute ischemic heart events such as myocardial infarction, angina, SCD, etc., are life-threatening conditions with high mortality and a major reason for performing forensic autopsies [84].

Among the main pathogenetic effects of cocaine on the cardiovascular system is the increased oxygen need of the myocardium (cocaine-induced hypertension) and, at the same time, the reduction in the myocardial oxygen supply through vasoconstriction of the epicardial coronary arteries [85]. For example, acute cocaine use enhances sympathomimetic activity by increasing catecholamine levels [86]. The basic mechanism includes: (i) increased α1-adrenergic stimulation, which promotes arterial vasoconstriction and increases heart rate, and (ii) increased heart rate and contractility due to enhanced β-adrenergic stimulation [87]. Simultaneously, cocaine acts as a local anesthetic, causing depression of the cardiovascular system [82]. Due to coronary vasoconstriction and the release of endothelin-1, the synthesis of endothelially produced nitric oxide is inhibited, and acetylcholine-induced vasorelaxation is impaired [49].

Acute use of low-dose cocaine increases heart rate, blood pressure, and myocardial contractility, which increases myocardial oxygen demand while reducing its supply. Compromised oxygen balance can lead to ischemia, angina [88]. The mechanisms show typical autopsy findings such as acute myocardial infarction, fibrosis replacement, and coronary thrombosis [83].

Cocaine and its significantly more toxic metabolite cocaethylene lead to ion channel blockade (the Na^+^ and K^+^ channels), prolonged QT interval, early postdepolarization, and ventricular tachyarrhythmia [89]. In combination with high levels of catecholamines, it can cause acidosis and electrolyte abnormalities, which increases the likelihood of cardiac arrhythmias [49]. In the event of sudden cardiac death due to cardiac arrhythmias and/or prolonged QT interval, no structural changes in the heart were observed [83].

### 6.2. Chronic Toxicity of Cocaine

#### 6.2.1. Thrombosis

In chronic cocaine abuse, activation of platelet aggregation and coagulation was observed by increasing fibrinogen production and decreasing the expression of antithrombin III and protein C [87]. Increased platelet aggregation, secretion of thrombogenic substances from the vascular endothelium, increased fibrinogen levels, and the von Willebrand factor were seen, which is associated with intravascular thrombosis in the coronary and peripheral arteries. Numerous studies showed that cocaine might alter blood viscosity, promoting thrombogenesis through increased interaction with tissue factor and tissue factor pathway inhibitor in endothelial and vascular smooth muscle cells [90].

#### 6.2.2. Atherosclerosis and Myocardial Infarction

Everyday cocaine use leads to endothelial damage, which promotes the early onset of coronary atherosclerosis, which is the leading cause of death in 28% of chronic cocaine users and is associated with intravascular thrombosis in the coronary and peripheral arteries. Scientific data show that cocaine activates platelet aggregation and stimulates the secretion of thrombogenic substances from the vascular endothelium, which causes acute or chronic myocardial ischemia [83].

In cases of acute non-cocaine MI, myocardial necrosis and accelerated thrombosis are usually associated with plaque fissure or hemorrhage rupture. In contrast, in cocaine abuse patients, no atherosclerotic lesions are usually seen [91]. Usually, myocardial infarction in these patients is characterized by inflammation of the arterial wall with many fibrous plaques rich in smooth muscle cells, lymphocytes, plasma cells, and mast cells. This finding is most commonly associated with increased myocardial oxygen consumption, coronary vasospasm, and prothrombotic status [92,93]. Kolodgie et al. reported cases of increased coronary atherosclerosis without plaque hemorrhage and elevated mast cell levels in subjects with cocaine-related thrombosis. Their study showed a high correlation between the number of adventitious mast cells and the degree of narrowing of the luminal cross-section in subjects with cocaine-initiated intracoronary thrombosis (r = 0.65), compared with subjects with sudden death due to thrombosis without evidence of cocaine use (r = 0.34) and [94].

#### 6.2.3. Left Ventricular Hypertrophy and Dilated Cardiomyopathy

Chronic cocaine abuse promotes myocardial hypertrophy and subsequent probable myocardial ischemia and/or arrhythmias [95]. Prolonged cocaine use increases end-diastolic pressure and end-left systolic end-volume, which is associated with left ventricular hypertrophy [96]; left ventricular failure [97]; decreased coronary blood flow; nonischemic myocardial depression; and dilated cardiomyopathy [98]. Numerous studied deaths from myocardial fibrosis among cocaine addicts reveal the presence of previous ischemic episodes. The so-called “silent ischemia”, along with myocardial fibrosis, can increase the risk of arrhythmias. There is a progressive accumulation of collagen fibers in the myocardial interstitium, which is due to cardiac hypertrophy, which occurs as a result of multiple high blood pressure episodes. This histological pattern more often affects the left ventricle, and it is called “cocaine-related cardiomyopathy” [99].

#### 6.2.4. Myocarditis

As a result of a hypersensitivity reaction to cocaine, local immune reactions may occur, leading to an inflammatory response in the myocardium, expressed in the presence of a mononuclear cell infiltrate. Such an inflammatory response in the myocardium and subsequent acute myocarditis were observed in 20% of cocaine-related toxic myocarditis deaths and were not dose-dependent [100]. Histological specimens examination of chronic cocaine users shows the presence of scattered foci of necrosis with loss of cardiac myofibrils, myocyte degeneration, and edema. Inflammatory cells that infiltrate necrotic tissue and remarkable mononuclear infiltration around cardiomyocytes can be seen [82].

#### 6.2.5. Endocardities

Compared to other drugs, cocaine use is associated with a higher incidence of endocarditis, which is probably due not only to the intravenous administration of the drug but also to a potential outbreak of bacterial adhesion through thrombus formation. In the general population of drug abusers, endocarditis is most common on the left side of the heart and affects the mitral or aortic valves. Seventy-six percent of cases of endocarditis after drug use are observed on the right side and include damage to the tricuspid valve (40–69%), aortic and mitral valves (20–30%), and more than one (5–10%) [101].

#### 6.2.6. Aortic Dissection

Cocaine use, and especially crack cocaine abuse, creates a rare but extremely fatal aortic dissection condition, probably due to decreased aortic elasticity and sudden and profound hypertension and tachycardia [102].

## 7. Free Radicals, Oxidative Stress, and Cocaine-Induced Cardio Toxicity

There is growing evidence of the influence of oxidative stress on the pathogenesis of cardiotoxicity in cocaine abusers. Oxidative stress (OS) occurs when the production of reactive oxygen and nitrogen exceeds the antioxidant defense systems of cells [102]. As a result of an imbalance between the generation and elimination of ROS and RNS, changes in redox homeostasis are achieved [103]. Although cocaine-induced toxicity is considered a brain disease and is one of the most studied neuropsychiatric disorders [104], OS-induced cocaine toxicity affects not only the brain but also the body as a whole. As a result, one of the most affected systems of oxidative damage is the cardiovascular system [20].

As a rule, abnormal levels of reactive oxygen species (ROS) and reactive nitrogen species (RNS) and the imbalance between anti- and pro-oxidants lead to lipid peroxidation, protein oxidation, DNA damage, and enzymatic dysfunctions, which can cause pathological changes in all organs and systems [103]. Cocaine administration leads to severe nitrosative/oxidative stress, mitochondrial dysfunction in the cardiomyocyte, and altered oxidative balance in the myocardium [104]. Due to high oxygen consumption, cardiac tissue is highly sensitive to hypoxia and lack of oxygen leads to the dysfunction of cardiomyocytes. Cellular hypoxia is associated with the generation of ROS such as •Ō_2_, H_2_O_2_, •OH and ONOO¯ generated by NADPH oxidase, mitochondrial electron transport, xanthine oxidase, etc. [105]. In heart failure or myocardial infarction (MI), cardiovascular oxidative stress leads to tissue damage by inducing vascular and myocardial inflammation, apoptosis, necrosis, hypertrophy, calcium dysregulation, and cardiovascular fibrosis [106].

### 7.1. The Basic Pathways of Cocaine Cardiotoxicity

Cocaine cardiotoxicity involves various direct and indirect mechanisms such as blockage of sodium, and potassium channels and altered calcium flow across the myocyte cell membranes, inhibition of reuptake, and increased levels of the neurotransmitters as dopamine and norepinephrine. Acute toxicity and drug dependence were associated with impaired energy and amino acid metabolism, increased oxidative stress levels, and altered dopamine neurotransmission. Moreover, of greater importance for the toxic manifestations and reactions of cocaine is the oxidative pathway involving electron transfer as a result of free radicals being generated [107]. Elevated levels of plasma and interstitial catecholamines and generated ROS and RNS cause prolonged adrenergic stress and a series of adverse effects with significant cardiotoxicity detrimental to the cardiovascular system [108]. Cocaine exposure leads to increased oxygen radical production, high levels of malondialdehyde MDA and nitrites in the prefrontal cortex and nucleus accumbens, oxidation of macromolecules, and oxidative damage to the brain [109]. Dopamine molecule oxidation by the enzyme monoamine oxidase produces superoxide radicals, hydrogen peroxide (H_2_O_2_), and reactive dopamine quinones [110]. Oxidative stress due to drug abuse is mediated primarily by dopamine, which is why the neurotransmitter pathway is considered an important source of ROS [111].

### 7.2. Enzymatic and Non-Enzymatic Pathways of ROS Cardiotoxicity

As additional sources of ROS with high cardiotoxicity, various aminochromes (adrenochromes, dopachrome, and noradrenochrome) are mentioned, which are produced during the catabolism of catecholamines [20]. As a result of oxidative stress and abundant ROS, catecholamines are transformed to aminochromes and, in particular, to highly toxic adrenochromes. It demonstrates direct cardiotoxicity to cardiomyocytes through disturbances in cellular Ca^2+^ homeostasis and disruption of oxidative phosphorylation, including an oxidation-reduction cycle with subsequent generation of ROS [19]. Adrenochrome, which is involved in this redox cycle, is responsible for the depletion of cellular antioxidants such as reduced ascorbate (AA) and glutathione (GSH), intracellular Ca^2+^ overload, lipid peroxidation, and cardiomyocyte damage by a ROS-dependent mechanism. Enzymatic and non-enzymatic degradation of catecholamines and stimulation of adrenergic receptors generate intracellular ROS in high concentrations [112]. An in vivo study in rats showed a significant increase in ROS levels and in the enzymes superoxide dismutase (SOD), glutathione peroxidase, and an increase in catalase activity (CAT) in the cerebral cortex and striatum [109,113]. The tripeptide glutathione (GSH, γ-glutamyl-cysteinyl-glycine) is the major intracellular antioxidant that participates as a cofactor of the four enzymes: glutathione peroxidase (GPx1), plasma GPx (GPx3), GPx2, and phospholipid hydroperoxide. They are involved in the conversion of H_2_O_2_ into water. Lipid peroxidation is a major cause of myocardial membrane phospholipid damage and leads to glutathione depletion in chronic cocaine use [114,115]. Macêdo et al. found that cocaine-induced death increased GSH levels in both single and multiple injections of cocaine in rats [106]. By ROS generation, cocaine severely compromised the antioxidant defense system in the heart by depleting non-enzymatic antioxidants such as glutathione in the myocardium. This thesis is confirmed by the study of Turillazzi et al. [108], who reported depletion of the antioxidant reserve expressed in the GSH/GSSG ratio, and decreased ascorbic acid (AA) levels and increased MDA concentrations in myocardial cells [105]. High levels of free radicals in cocaine intoxication can lead to a sudden compensatory increase in the activity of endogenous enzyme systems, whose role is to remove free radicals and reduce levels of oxidative stress. Important sources of ROS and RNS include enzymatic reactions involving cytochrome P450 enzymes, NAD(P)H oxidase (NOX), myeloperoxidase, and eosinophil peroxidase [20]. For example, neutrophil type NAD(P)H oxidases are a major ROS source in cardiovascular cells and are involved in the production of superoxide radical (•Ō_2_) in cardiac myocytes in cases of hypoxia or myocardial infarction and play an important role in angiotensin II-induced cardiac hypertrophy [116].

### 7.3. Mitochondrial Dysfunction and Nitrosative/Oxidative Stress

In recent years, there have been increasing scientific reports of cocaine-induced cardiac dysfunction as a result of nitrosative/oxidative stress after cocaine use and mitochondrial dysfunction as a result of oxidative damage to cellular structures [17,18,117,118]. Cocaine directly inhibits the mitochondrial electron transport chain by increasing intramitochondrial Ca^2+^ overload and depleting adenosine triphosphate (ATP) production. High levels of catecholamines disrupt calcium homeostasis and increase the activity of NAD(P)H oxidase and xanthine oxidase. As a result, additional amounts of ROS and RNS are generated in the mitochondria, which cause mitochondrial dysfunction. High ROS levels damage cellular macromolecules (DNA, lipids, and proteins) and adversely affect myocardial calcium function, causing arrhythmias, increasing cardiac remodeling, and leading to hypertension, necrosis, and apoptosis [47]. Lipid peroxidation is a major cause of myocardial membrane phospholipid damage and leads to glutathione depletion in chronic cocaine use [119]. Calcium and OS overload cause cardiomyocyte death in both the apoptotic and necrotic pathways. In cell line studies, ROS was found to have a primarily direct effect on cardiac cells by activating mitogen-activated protein kinase (MAPK) and NOX2. These results in acute myocardial oxidative stress, oxidative damage to cardiomyocytes, and cell death in a mouse model [120,121]. The first mechanism is the activation of the MAPK-beta-adrenergic receptor after calcium overload and subsequent phosphorylation of a multitude of calcium-containing cyclic proteins. The second mechanism is based on the redox cycle and takes place in the mitochondria [122]. It is also known that high levels of oxidative stress can cause mitochondrial transition pore opening, leading to the generation of abnormal ROS levels and so-called ROS-induced ROS release. This process significantly increases the production of ROS and is considered to be a major mechanism of pathological changes in the entire cardiovascular system, such as cardiac ischemia-reperfusion and subsequent acute myocardial infarction [122,123]. Numerous studies considered the relationship between oxidative stress and free radical generation in oxidative myocardial damage, and special attention was paid to the compromise of the antioxidant defense system of the heart in the pathogenesis of cardiotoxicity in the administration of illicit substances [121,124].

### 7.4. ROS Production from Endothelial Cells

The mechanism of cocaine toxicity involves the expression of enzymes and compromise of mitochondrial function through direct or indirect action of cocaine on endothelial cells. Mitochondria are known to be the main source of reactive oxygen, not only in cardiomyocytes but also in endothelial cells [125]. Due to the presence of enzymes such as XO, NOS, and mitochondrial MAO and NAPDH oxidase, endothelial cells can also generate ROS [111]. In endothelial cells with decreased xanthine dehydrogenase (XDH) levels, increased production of superoxide anion radical is induced [126]. Due to the strong cardiotoxicity of the cocaine molecule as a result of increased levels of oxidative stress, the antioxidant defense system in the cells is disrupted, causing oxidative damage to cell structures and myocardial tissue as a whole [127].

### 7.5. Nitroxide Radicals as Detector of Oxidative Stress in Human Body

Nitroxides are a class of stable, cyclic, redox-sensitive paramagnetic species that have an unpaired electron (NO•). They are divided into two major structural groups, pyrrolidine (five-membered ring) and piperidine (six-membered ring) derivatives. By chemically modifying the functional groups in the ring, their chemical and biological behavior and biodistribution can be controlled [128]. Due to their paramagnetic nature, nitroxides are used as redox sensors in electron paramagnetic resonance spectroscopy (EPR) and as contrast agents in nuclear magnetic resonance imaging (MRI) to detect redox changes in biological systems [129,130]. They cross the cell membrane and participate in various redox reactions involving heme proteins, thiols, and ROS [129,131]. In the presence of endogenous oxidants (ROS, RNS), through electron transfer reactions, paramagnetic nitroxides are converted into their diamagnetic form hydroxylamine (N-OH) or oxoammonium cation (Figure 3), which makes them suitable tools for monitoring redox changes in cells, tissues, and organs. In general, the metabolic rates (bioreduction) and biological effects of nitroxides are mainly determined by their structure and functional groups. For example, in a tumor microenvironment, the pyridine nitroxide 4-hydroxy-2,2,6,6-tetramethylpiperidin-1-oxyl (Tempol) is reduced to its hydroxylamine form, 11 times faster than 3-Carbamoyl-PROXYL radical [132]. The reduction in nitroxide to hydroxylamine results in a decrease in the EPR signal, and the rate of reduction in the spin probe may be increased or decreased in conditions characterized by abnormal oxidative stress [133]. Clinical applications of nitroxides are mainly in monitoring and protective effects in conditions characterized by high levels of free radicals such as cancer [134], inflammation [135,136], neurodegenerative diseases [16], and others. Tempol is known to have therapeutic potential and is used in diseases associated with high oxidative stress. It acts as an antioxidant by suppressing the formation of hydroxyl radicals by inhibiting the Fenton reaction [108,137].

It is known that Tempol has therapeutic potential and finds application in medical conditions with a high level of oxidative stress. It is characterized by dose-dependent effects and acts as an antioxidant by inhibiting the Fenton reaction [108].

Although the role of OS in the pathogenesis of cardiotoxicity in cocaine intoxication is well known, to date, the potential use of nitroxide radicals as redox-sensitive detectors to determine OS levels in cocaine-induced cardiotoxicity has not been considered. In the context of their redox properties, they can find wide application in determining the severity and consequences of acute and chronic drug intoxication.

## 8. Conclusions

Cocaine can cause acute complications, arrhythmias, myocardial infarction and may be associated with vascular disease, heart disease, and coronary heart disease in case of chronic intoxication after prolonged use of the drug. In general, cocaine abuse leads to serious pathological changes and irreversible damage to the entire cardiovascular system. The picture is often complicated by the fact that people who abuse drugs often take them with other drugs and or alcohol, the combination of which can put the user in life-threatening situations. High awareness of the harmful effects of drug use, early diagnosis, knowledge of life-threatening cardiovascular conditions, and timely medical care are the main points in the successful treatment and saving of human life. Globally, deaths from stimulant use are skyrocketing. This requires additional preclinical and clinical studies to allow the development of a methodology for rapid qualitative and quantitative analysis of individuals with cocaine intoxication. The superoxide radical, the hydroxyl radical, the hydrogen peroxide, the alkyl peroxides, and others, together with the depletion of glutathione, are the main reasons for causing irreversible pathological and morphological changes in the cells, tissues, and organs. As cocaine use leads to the generation of free radicals, future methods for the analysis of acute and chronic drug intoxication should be based on changes in redox status and appropriate spin probes.

## Data Availability

Data available in a publicly accessible repository.

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
