# Peer review of "Oxidative Stress and Cocaine Intoxication as Start Points in the Pathology of Cocaine-Induced Cardiotoxicity"

_toxics, 2021, doi:10.3390/toxics9120317_

Round 1

Reviewer 1 Report

this review does not include many autopsy studies of cardiac findings in deaths attributed to cocaine, ie. infarcts, precocious atheroma, chronic damage, left ventricular hypertrophy aortic dissection myocarditis, endocarditis. only  ref 52 mentions cardiac pathology and autopies. Can you illustrate the effects seen in the heart with cocaine found at autopsy and mention more studies not just case reports . Distinguish between the acute and chronic effects on the heart found at autopsy . How do you help the pathologist to attribute the death to cocaine use?  

Author Response

Dear reviewer,

Thank you very for helping us to improve our manuscript.

According to your remarks in relation to our manuscript toxics-1443568 titled: "Oxidative stress and cocaine intoxication as start points in the pathology of cocaine -induced cardiotoxicity" Ekaterina Georgieva, Yanka Karamalakova, Radostina Miteva, Hristo Abrashev, Galina Nikolova we have made the corresponding corrections and additions:

Point 1

Reviewer: “this review does not include many autopsy studies of cardiac findings in deaths attributed to cocaine, ie. infarcts, precocious atheroma, chronic damage, left ventricular hypertrophy aortic dissection myocarditis, endocarditis. nly ref 52 mentions cardiac pathology and autopies. Can you illustrate the effects seen in the heart with cocaine found at autopsy and mention more studies not just case reports. Distinguish between the acute and chronic effects on the heart found at autopsy. How do you help the pathologist to attribute the death to cocaine use?”

Response 1: We agree with the reviewer's opinion and therefore we have added a section describing the damage caused by cocaine in a little more detail.

ACUTE AND CHRONIC EFFECTS OF COCAINE CARDIOVASCULAR TOXICITY AND PATHOLOGICAL CHANGES ON THE CARDIOVASCULAR SYSTEM.

In general, the number of cardiovascular events following cocaine abuse is increasing dramatically, but distinguishing the effects and determining the consequences in humans is troublesome, as there are a number of factors such as differences in administration, doses, mixing cocaine with other drugs (alcohol, caffeine and amphetamines), the presence of contaminants of the probes (procainamide, quinidine, antihistamines), etc. Numerous and varied pathological changes are known after acute and chronic cocaine abuse, such as myocardial infarction, myocarditis, catecholamine-induced cardiomyopathy, chronic cardiomyopathy, etc., but their clinical manifestations overlap to a large extent [81]. ​In order to determine the cause and mechanism of sudden cardiac death (SCD), it is necessary to a detailed forensic investigation to be done, which will allow the understanding of the intimate mechanism and whether the disease is due to acute or chronic cocaine use. Toxicological analyzes would provide clarity regarding the toxic agent, while mac-roscopic and histopathological examination would make it possible to establish the mechanism and role of the intoxicant that led to the lethal outcome. Mechanisms associ-ated with an increased risk of SCD after acute or chronic cocaine use include coronary vasoconstriction, accelerated atherosclerosis, increased thrombogenesis, left ventricular hypertrophy, increased myocardium oxygen needs, acute myocardial infarction, etc. [82].

Acute toxicity of cocaine. Myocardial infarction, arrhythmias, sudden cardiac death

Cocaine-induced cardiotoxicity can cause a variety of structural and functional damage to cardiac tissue [48]. Acute ischemic heart events such as myocardial infarction, angina, SCD, etc., are life-threatening conditions with high mortality and a major reason for performing forensic autopsies [83].

Among the main pathogenetic effects of cocaine on the cardiovascular system is the increased oxygen need of the myocardium (cocaine-induced hypertension) and at the same time the reduction of myocardial oxygen supply through vasoconstriction of the epicardial coronary arteries [84]. For example, acute cocaine use enhances sympathomimetic activity by increasing catecholamine levels [85]. The basic mechanism includes: i) increased α1-adrenergic stimulation, which promotes arterial vasoconstriction and increases heart rate; and (ii) increased heart rate and contractility due to enhanced β-adrenergic stimulation [86]. Simultaneously, cocaine acts as a local anesthetic, causing depression of the cardiovascular system [81]. Due to coronary vasoconstriction and the release of endothelin-1, the synthesis of endothelially produced nitric oxide is inhibited and acetylcholine-induced vasorelaxation is impaired [85].

Acute use of low-dose cocaine increases heart rate, blood pressure, and myocardial contractility, which increases myocardial oxygen demand, while reducing its supply. Compromised oxygen balance can lead to ischemia, angina [87]. The mechanisms show typical autopsy findings such as acute myocardial infarction, fibrosis replacement and coronary thrombosis [82].

Cocaine and its significantly more toxic metabolite cocaethylene lead to ion channel blockade (the Na+ and K+ channels), prolonged QT interval, early postdepolarization, and ventricular tachyarrhythmia [88]. In combination with high levels of catecholamines, cause acidosis and electrolyte abnormalities, which increases the likelihood of cardiac arrhythmias [85]. In the event of sudden cardiac death due to cardiac arrhythmias and/or prolonged QT interval, no structural changes in the heart were observed [82].

Hronic toxicity of cocaine      

Thrombosis

 In chronic cocaine abuse, activation of platelet aggregation and coagulation has been observed by increasing fibrinogen production and decreasing the expression of antithrombin III and protein C [86]. Increased platelet aggregation, secretion of thrombogenic substances from the vascular endothelium, increased fibrinogen levels and von Willebrand factor have been seen, which is associated with intravascular thrombosis in the coronary and peripheral arteries. Numerous studies show that cocaine may alter blood viscosity, promoting thrombogenesis through increased interaction with tissue factor and tissue factor pathway inhibitor in endothelial and vascular smooth muscle cells [89].

Atherosclerosis and myocardial infarction

Everyday cocaine use leads to endothelial damage, which promotes the early onset of coronary atherosclerosis, which is the leading cause of death in 28% of chronic cocaine users and is associated with intravascular thrombosis in the coronary and peripheral arteries. Scientific data show that cocaine activates platelet aggregation and stimulates the secretion of thrombogenic substances from the vascular endothelium, which causes acute or chronic myocardial ischemia [82].

In cases of acute non-cocaine MI, myocardial necrosis and accelerated thrombosis are usually associated with plaque fissure or hemorrhage rupture. In contrast, in cocaine abuse patients no atherosclerotic lesions are usually seen. Usually, myocardial infarction in these patients is characterized by inflammation of the arterial wall with many fibrous plaques rich in smooth muscle cells, lymphocytes, plasma cells and mast cells. This finding is most commonly associated with increased myocardial oxygen consumption, coronary vasospasm, and prothrombotic status [91, 92]. Kolodgie et al. reported cases of increased coronary atherosclerosis without plaque hemorrhage and elevated mast cell levels in subjects with cocaine-related thrombosis. Their study showed a high correlation between the number of adventitious mast cells and the degree of narrowing of the luminal cross-section in subjects with cocaine-initiated intracoronary thrombosis (r = 0.65), compared with subjects with sudden death due to thrombosis without evidence of cocaine use (r = 0.34) and [93].

Left ventricular hypertrophy and dilated cardiomyopathy

Chronic cocaine abuse promotes myocardial hypertrophy and subsequent probable myocardial ischemia and/or arrhythmias [94]. Prolonged cocaine use increases end-diastolic pressure and end-left systolic end-volume, which is associated with left ventricular hypertrophy [95], left ventricular failure [96], decreased coronary blood flow, nonischemic myocardial depression and dilated cardiomyopathy [97]. Numerous studied deaths from myocardial fibrosis among cocaine addicts reveal the presence of previous ischemic episodes. So-called "silent ischemia", along with myocardial fibrosis, can increase the risk of arrhythmias. There is a progressive accumulation of collagen fibers in the myocardial interstitium, which is due to cardiac hypertrophy, which occurrs as a result of multiple high blood pressure episodes. This histological pattern, more often affects the left ventricle and it is called "cocaine-related cardiomyopathy" [98].

Myocarditis

As a result of a hypersensitivity reaction to cocaine, local immune reactions may occur, leading to an inflammatory response in the myocardium, expressed in the presence of a mononuclear cell infiltrate. Such an inflammatory response in the myocardium and subsequent acute myocarditis were observed in 20% of cocaine-related toxic myocarditis deaths and were not dose-dependent [99]. Histological specimens examination of chronic cocaine users show the presence of scattered foci of necrosis with loss of cardiac myofibrils, myocyte degeneration and edema. Inflammatory cells that infiltrate necrotic tissue and remarkable mononuclear infiltration around cardiomyocytes can be seen [81].

Endocardities

Compared to other drugs, cocaine use is associated with a higher incidence of endocarditis, which is probably due not only to the intravenous administration of the drug, but also to a potential outbreak of bacterial adhesion through thrombus formation. In the general population out of drug abusers, endocarditis is most common on the left side of the heart and affects the mitral or aortic valves. 76% of cases of endocarditis after drug use are observed on the right side and include damage to the tricuspid valve (40% -69%), aortic and mitral valves (20% -30%,) and more than one (5% - 10%) [100]

Aortic dissection

Cocaine use, and especially crack cocaine abuse, creates a rare but extremely fatal aortic dissection condition, probably due to decreased aortic elasticity and sudden and profound hypertension and tachycardia [101].

Reviewer 2 Report

# OXIDATIVE STRESS AND COCAINE INTOXICATION AS START POINTS IN THE PATHOLOGY OF COCAINE -INDUCED CARDIOTOXICITY
The presented manuscript by Georgieva et al. reviews literature subjected to cocaine intoxications, the concerning cardiotoxicity and the role of reactive oxygen species. The authors intend to provide a new viewpoint in assessing cocaine intoxications by not only focusing on cerebral complications of cocaine use but also the cardiotoxicity. They provide a general overview of the mode of action of cocain, first regarding its cerebral effects and in the course of the review also regarding its cardiotoxic effects.
The manuscript needs several passages to be rephrased since the used language is misleading, especially when presenting statistical numbers, which are often inapproprietly generalized. I pointed out several passages where this applies and noted some additional minor changes, that need to be addressed before this manuscript is suitable for publishing.
Please read the following nomenclature of pages and lines as follows: P5L5 = Page 5, Line 5

## Abstract
P1L16: Are psychomotor stimulants the most common cause of death regarding all possible causes of death or does this statement related to deaths where drugs of abuse are involved? I am not an epidemiologist, but I believe cardiovascular events and strokes are more common.

P1L17: I believe this sentences got chopped up somehow. And again, I believe this statement can't be made in such a general way.

P1L23: Not sure what your intention is with this statement. Please also note my comment regarding P1L63.

## Introduction
P1L37: This is again a bit chopped up.

P1L49-L56: I find this passage is essentially repeating the same statement over and over. Maybe it helps, if you try to describe the inflicted damage of cocaine and its metabolite slightly more in detail.

P1L63: I believe you want to say that this review might provide an overview in assessing acute and chronic cocaine toxicity by focusing on the published literature regarding OS levels. Please rephrase since your current wording implies that you performed OS level studies.

P4L178: This statement is slightly misleading. A molecule is not cardiotoxic by simply effecting a lot of receptors. Furthermore, "entire systems" is a bit of a spongy expression.

P5L222: I think it is appropriate to use numbers for "forty-three". But I am again surprised about the statement, since I cannot believe that cocain consumption is that widespread that 43% of all emergency room visits are caused by cocaine use. Is this number related to emergency room visits among general drugs of abuse related visits?

P6L240: "Norcocaine" instead of "Norcacaine".

Author Response

Dear reviewer,

Thank you very for helping us to improve our manuscript.

According to your remarks in relation to our manuscript toxics-1443568 titled: "Oxidative stress and cocaine intoxication as start points in the pathology of cocaine -induced cardiotoxicity" Ekaterina Georgieva, Yanka Karamalakova, Radostina Miteva, Hristo Abrashev, Galina Nikolova we have made the corresponding corrections and additions:

Reviewer: # OXIDATIVE STRESS AND COCAINE INTOXICATION AS START POINTS IN THE PATHOLOGY OF COCAINE -INDUCED CARDIOTOXICITY
The presented manuscript by Georgieva et al. reviews literature subjected to cocaine intoxications, the concerning cardiotoxicity and the role of reactive oxygen species. The authors intend to provide a new viewpoint in assessing cocaine intoxications by not only focusing on cerebral complications of cocaine use but also the cardiotoxicity. They provide a general overview of the mode of action of cocaine, first regarding its cerebral effects and in the course of the review also regarding its cardiotoxic effects. The manuscript needs several passages to be rephrased since the used language is misleading, especially when presenting statistical numbers, which are often inapproprietly generalized. I pointed out several passages where this applies and noted some additional minor changes,that need to be addressed before this manuscript is suitable for publishing. Please read the following nomenclature of pages and lines as follows: P5L5 = Page 5, Line 5

Point 1

## Abstract
P1L16: Are psychomotor stimulants the most common cause of death regarding all possible causes of death or does this statement related to deaths where drugs of abuse are involved? I am not an epidemiologist, but I believe cardiovascular events and strokes are more common.

Response 1: Globally, their use reaching epidemiological proportions and is one of the most common causes of death after the administration of drugs.

Point 2

P1L17: I believe these sentences got chopped up somehow. And again, I believe this statement can't be made in such a general way.

Response 2: The use of illicit drugs has negative effects on the cardiovascular system and is one of the causes of serious cardiovascular pathologies, ranging from abnormal heart rhythms to heart attacks and sudden cardiac death.

Point 3/ P1L23: Not sure what your intention is with this statement. Please also note my comment regarding P1L63.

Response 3: The aim of the present review is to assess the acute and chronic cocaine toxicity with focusing on the published literature regarding oxidative stress levels.

## Introduction
Point 4/ P1L37: This is again a bit chopped up.

Response 4: We agree with the reviewer that the sentence is redundant and therefore removed it.

Point 5/ P1L49-L56: I find this passage is essentially repeating the same statement over and over. Maybe it helps, if you try to describe the inflicted damage of cocaine and its metabolite slightly more in detail.

Response 5: We agree with the reviewer's opinion and therefore we have added a section describing the damage caused by cocaine and its metabolite in a little more detail.

Acute toxicity of cocaine. Myocardial infarction, arrhythmias, sudden cardiac death

Cocaine-induced cardiotoxicity can cause a variety of structural and functional damage to cardiac tissue [48]. Acute ischemic heart events such as myocardial infarction, angina, SCD, etc., are life-threatening conditions with high mortality and a major reason for performing forensic autopsies [83].

Among the main pathogenetic effects of cocaine on the cardiovascular system is the increased oxygen need of the myocardium (cocaine-induced hypertension) and at the same time the reduction of myocardial oxygen supply through vasoconstriction of the epicardial coronary arteries [84]. For example, acute cocaine use enhances sympathomimetic activity by increasing catecholamine levels [85]. The basic mechanism includes: i) increased α1-adrenergic stimulation, which promotes arterial vasoconstriction and increases heart rate; and (ii) increased heart rate and contractility due to enhanced β-adrenergic stimulation [86]. Simultaneously, cocaine acts as a local anesthetic, causing depression of the cardiovascular system [81]. Due to coronary vasoconstriction and the release of endothelin-1, the synthesis of endothelially produced nitric oxide is inhibited and acetylcholine-induced vasorelaxation is impaired [85].

Acute use of low-dose cocaine increases heart rate, blood pressure, and myocardial contractility, which increases myocardial oxygen demand, while reducing its supply. Compromised oxygen balance can lead to ischemia, angina [87]. The mechanisms show typical autopsy findings such as acute myocardial infarction, fibrosis replacement and coronary thrombosis [82].

Cocaine and its significantly more toxic metabolite cocaethylene lead to ion channel blockade (the Na+ and K+ channels), prolonged QT interval, early postdepolarization, and ventricular tachyarrhythmia [88]. In combination with high levels of catecholamines, cause acidosis and electrolyte abnormalities, which increases the likelihood of cardiac arrhythmias [85]. In the event of sudden cardiac death due to cardiac arrhythmias and/or prolonged QT interval, no structural changes in the heart were observed [82].

Hronic toxicity of cocaine      

Thrombosis

 In chronic cocaine abuse, activation of platelet aggregation and coagulation has been observed by increasing fibrinogen production and decreasing the expression of antithrombin III and protein C [86]. Increased platelet aggregation, secretion of thrombogenic substances from the vascular endothelium, increased fibrinogen levels and von Willebrand factor have been seen, which is associated with intravascular thrombosis in the coronary and peripheral arteries. Numerous studies show that cocaine may alter blood viscosity, promoting thrombogenesis through increased interaction with tissue factor and tissue factor pathway inhibitor in endothelial and vascular smooth muscle cells [89].

Atherosclerosis and myocardial infarction

Everyday cocaine use leads to endothelial damage, which promotes the early onset of coronary atherosclerosis, which is the leading cause of death in 28% of chronic cocaine users and is associated with intravascular thrombosis in the coronary and peripheral arteries. Scientific data show that cocaine activates platelet aggregation and stimulates the secretion of thrombogenic substances from the vascular endothelium, which causes acute or chronic myocardial ischemia [82].

In cases of acute non-cocaine MI, myocardial necrosis and accelerated thrombosis are usually associated with plaque fissure or hemorrhage rupture. In contrast, in cocaine abuse patients no atherosclerotic lesions are usually seen. Usually, myocardial infarction in these patients is characterized by inflammation of the arterial wall with many fibrous plaques rich in smooth muscle cells, lymphocytes, plasma cells and mast cells. This finding is most commonly associated with increased myocardial oxygen consumption, coronary vasospasm, and prothrombotic status [91, 92]. Kolodgie et al. reported cases of increased coronary atherosclerosis without plaque hemorrhage and elevated mast cell levels in subjects with cocaine-related thrombosis. Their study showed a high correlation between the number of adventitious mast cells and the degree of narrowing of the luminal cross-section in subjects with cocaine-initiated intracoronary thrombosis (r = 0.65), compared with subjects with sudden death due to thrombosis without evidence of cocaine use (r = 0.34) and [93].

Left ventricular hypertrophy and dilated cardiomyopathy

Chronic cocaine abuse promotes myocardial hypertrophy and subsequent probable myocardial ischemia and/or arrhythmias [94]. Prolonged cocaine use increases end-diastolic pressure and end-left systolic end-volume, which is associated with left ventricular hypertrophy [95], left ventricular failure [96], decreased coronary blood flow, nonischemic myocardial depression and dilated cardiomyopathy [97]. Numerous studied deaths from myocardial fibrosis among cocaine addicts reveal the presence of previous ischemic episodes. So-called "silent ischemia", along with myocardial fibrosis, can increase the risk of arrhythmias. There is a progressive accumulation of collagen fibers in the myocardial interstitium, which is due to cardiac hypertrophy, which occurrs as a result of multiple high blood pressure episodes. This histological pattern, more often affects the left ventricle and it is called "cocaine-related cardiomyopathy" [98].

Myocarditis

As a result of a hypersensitivity reaction to cocaine, local immune reactions may occur, leading to an inflammatory response in the myocardium, expressed in the presence of a mononuclear cell infiltrate. Such an inflammatory response in the myocardium and subsequent acute myocarditis were observed in 20% of cocaine-related toxic myocarditis deaths and were not dose-dependent [99]. Histological specimens examination of chronic cocaine users show the presence of scattered foci of necrosis with loss of cardiac myofibrils, myocyte degeneration and edema. Inflammatory cells that infiltrate necrotic tissue and remarkable mononuclear infiltration around cardiomyocytes can be seen [81].

Endocardities

Compared to other drugs, cocaine use is associated with a higher incidence of endocarditis, which is probably due not only to the intravenous administration of the drug, but also to a potential outbreak of bacterial adhesion through thrombus formation. In the general population out of drug abusers, endocarditis is most common on the left side of the heart and affects the mitral or aortic valves. 76% of cases of endocarditis after drug use are observed on the right side and include damage to the tricuspid valve (40% -69%), aortic and mitral valves (20% -30%,) and more than one (5% - 10%) [100]

Aortic dissection

Cocaine use, and especially crack cocaine abuse, creates a rare but extremely fatal aortic dissection condition, probably due to decreased aortic elasticity and sudden and profound hypertension and tachycardia [101].

Point 6/ P1L63:  I believe you want to say that this review might provide an overview in assessing acute and chronic cocaine toxicity by focusing on the published literature regarding OS levels. Please rephrase since your current wording implies that you performed OS level studies.

Response 6: Cocaine exerts its toxicity on the human body by generating ROS, such as hydrogen peroxide, hydroperoxides, alkyl peroxides, superoxide, hydroxyl, and others [20]. According to the data, ROS production and oxidative damage are considered as a significant factor in the cocaine- induced cardiotoxicity [21, 22] and pathogenesis of cardiovascular desiese [23].

Point 7/ P4L178: This statement is slightly misleading. A molecule is not cardiotoxic by simply effecting a lot of receptors. Furthermore, "entire systems" is a bit of a spongy expression.

Response 7: Cocaine has very powerful direct effects on cell membranes by blocking sodium channel activity, so cocaine intoxication is a common cause of various cardiovascular events.

Point 8/ P5L222: I think it is appropriate to use numbers for "forty-three". But I am again surprised about the statement, since I cannot believe that cocaine consumption is that widespread that 43% of all emergency room visits are caused by cocaine use. Is this number related to emergency room visits among general drugs of abuse related visits?

Response 8: About 43% of emergency department visits after drug intoxication are due to cocaine use, with the highest percentage of men between 35-44 years of age [67]. Mortality associated with cocaine use is also common in 30-year-old men and is most common at home and on weekends.

Point 9/ P6L240: "Norcocaine" instead of "Norcacaine".

Response 9: Norcocaine

Round 2

Reviewer 1 Report

much improved with detailed analysis of cardiac effects of cocaine